

# Cells tile a flat plane by controlling geometries during morphogenesis of *Pyropia* thalli

Kai Xu, Yan Xu, Dehua Ji, Ting Chen, Changsheng Chen and Chaotian Xie

Fisheries College, Jimei University, Xiamen, Fujian, China
Key Laboratory of Healthy Mariculture for the East China Sea, Ministry of Agriculture, Xiamen, Fujian, China

## ABSTRACT

**Background**. *Pyropia haitanensis* thalli, which are made of a single layer of polygonal cells, are a perfect model for studying the morphogenesis of multi-celled organisms because their cell proliferation process is an excellent example of the manner in which cells control their geometry to create a two-dimensional plane.

**Methods**. Cellular geometries of thalli at different stages of growth revealed by light microscope analysis.

**Results**. This study showed the cell division transect the middle of the selected paired-sides to divide the cell into two equal portions, thus resulting in cell sides ≥4 and keeping the average number of cell sides at approximately six even as the thallus continued to grow, such that more than 90% of the cells in thalli longer than 0.08 cm had 5–7 sides. However, cell division could not fully explain the distributions of intracellular angles. Results showed that cell-division-associated fast reorientation of cell sides and cell divisions together caused 60% of the inner angles of cells from longer thalli to range from 100–140°. These results indicate that cells prefer to form regular polygons.

**Conclusions**. This study suggests that appropriate cell-packing geometries maintained by cell division and reorientation of cell walls can keep the cells bordering each other closely, without gaps.

Corresponding author
Chaotian Xie, ctxie@jmu.edu.cn

## INTRODUCTION

*Pyropia* is an intertidal red algae. Its edible portion forms during the thallus stage and has an annual production worth about 1.3 billion USD (*Blouin et al., 2011*). The thallus is a membranous sheet in a lanceolate shape made of one or two layers of cells. Two of the most economically important cultured species, *Pyropia yezoensis* and *P. haitanensis*, only contain one layer of cells. During the thallus stage, *P. haitanensis* thalli can take on three morphologies in sequence: single-celled conchospores (point), linearly ordered groups of 4–10 cells (line), and a membranous sheet (plane). The cell proliferation during morphogenesis of *P. haitanensis* thalli is essentially two-dimensional (2D) expansion on a plane. The specific geometries make *P. haitanensis* a simple but valuable model organism for the study of the morphogenesis of multi-celled organisms. Although the cell-packing

geometries keep changing due to cell growth and division, most of the cells could be considered convex polygons with a small number of spherical cells at the base. The morphogenesis of thalli features cells that border each other closely with no empty spaces or gaps. The mechanisms underlying this feature are equivalent to a mathematical question regarding how convex polygons tile or tessellate in regular patterns on 2D planes. The geometric patterns of *Pyropia* cells also follow the mathematical laws and must be tightly controlled, but the patterns and underlying control mechanisms are poorly understood.

Three laws were here generalized for the analysis of general topological properties of 2D tessellation: Euler's law (faces − edges + vertex = 1), Lewis' law (the relationship between mean area of a convex n-sided cell and n) and Aboav–Weaire law (*Aboav, 1980*) (the relationship between the mean number of sides of neighboring cells of a convex n-sided cell and n) (*Aboav, 1980*; *Lewis, 1928*; *Sanchez-Gutierrez et al., 2016*; *Weaire & Rivier, 1984*). Two basic mathematical generalizations were found to underlie the tessellations in which only one kind of polygon was used to tile a flat plane (*Grünbaum & Shephard, 1987*; *Lord, 2016*): 1. Any kind of polygon with more than 6 sides would be unable to form a close tile pattern on a flat plane; 2. To date, 15 irregular pentagons, 16 hexagons (including regular hexagon) and all triangles and quadrilaterals have been confirmed to be able to form close tile patterns on flat planes. However, the tessellation of *Pyropia* thalli is the tiling of a flat plane using more than one kind of polygon due to growth and cell division changing the cell-packing geometries. Conserved distribution of cellular polygons has been observed in many proliferating tissues. It generally features a predominance of hexagonal cells and an average of 6 sides, and it is considered as a mathematically determined consequence of cell proliferation (*Gibson et al., 2006*; *Graustein, 1931*; *Lewis, 1926*; *Lewis, 1928*). However, a recent study reported that many different natural tissues have very different distributions of polygons (*Sanchez-Gutierrez et al., 2016*).

During the past few decades, few studies have focused on cell-packing geometry, mostly by studying the epithelial cells of *Drosophila* wings, which can transition from irregular arrangements to hexagonal patterns before hair formation (*Classen et al., 2005*; *Farhadifar et al., 2007*; *Gibson et al., 2006*). It is still under debate whether the cellular geometry is achieved by cell division (*Gibson et al., 2006*) or cell rearrangement mediated by the physical properties of the cells (*Classen et al., 2005*; *Farhadifar et al., 2007*). A recent work by *Sanchez-Gutierrez et al. (2016)* has shed light on the mechanism that drives packed tissue organization. Their paper demonstrated an interesting finding, that sections of muscle fiber (a natural non-proliferative tissue) can be enriched in hexagons. They found that a physical constraint that induced by the balance of forces between cells in non-proliferative and proliferative tissues can cause all packed tissues to be arranged in a manner similar to Voronoi diagrams. Thus, *Sanchez-Gutierrez et al. (2016)* suggested that conserved polygon distribution does not necessarily need the proliferation mechanisms.

In a model study, *Patel et al. (2009)* found that strong division symmetry is required to reproduce the predominance of hexagonal cells, which indicates that the orientation of the mitotic cleavage plane was not random. The surface energy and tension of cell membrane were mainly determined by the shape of protoplasm; these biophysical properties were found to change the physiological performance and gene expression (*Chen, 2008*; *Ingber,*

*Wang & Stamenovic, 2014*). In this way, the interior angles of the polygonal cells not only determined the bordering patterns of cells but could also change the force balance between cells. Thus, if the initial cellular geometries of two daughter cells were determined solely by the original geometries of mother cell and the orientation of the cleavage plane, then the force balance would be disturbed by formation of the interior angles of daughter cells. In this study, to understand the dynamic processes and mechanisms underlying cell-packing geometries during morphogenesis of *P. haitanensis* thalli, samples collected during different stages of growth were used for analysis.

## MATERIALS AND METHODS

The materials used in this study were three strains of *P. haitanensis* present: Z-61 (red-brown), G-2 (green), and O-9 (orange). These strains were selected by the Laboratory of Germplasm Improvements and Applications of *Pyropia* in Jimei University, Fujian Province, China. Ripened free-living conchocelis of all three strains were cultured in natural seawater at 21 °C and 50 $\mu$mol photons m$^{-2}$ s$^{-1}$ (12:12, L/D cycle). The culture media were natural seawater enriched with nutrients according to Provasoli's enrichment solution (PES) and refreshed every two days. The culture media bubbled continuously using filter-sterilized air to promote the liberation of conchospores. Thalli arose from meiosis after conchospore germination. Healthy thalli were investigated and photographed under a microscope (Nikon U20).

Cells located at the base of the thallus were excluded from the following analysis because they were too circular in form. Only cells completely surrounded by other cells were selected. Because the thalli were considered a 2D plane in this study and all the convex polygon shaped cells of the thalli bordered each other closely, then the geometries of the polygonal cells were quantified to analyze the tessellation of the 2D thalli. The cell sides were counted, and the measurements of cell area (polygon area), side length, and intracellular angles (interior angles of the polygonal cell) were made using AmScope Toupview 3.0. If the length of a boundary exceeded 1 $\mu$m (cell wall $\approx$ 0.5–1 $\mu$m), then that boundary was considered a side. To help count the sides of the cells, photos were treated with two color effects (black and white and/or inversion) using Google Picasa 3.0 software.

Cell division was identified using time-series observation. The daughter cells were assumed to have the same height, and the size ratio of daughter cells was calculated as SR $= $ As $\div$ Ab, where As and Ab are area of the smaller and larger daughter cells, respectively. Cell division section was calculated as DS $= |$S1 $\div$ (S1 $+$ S2) $- 0.5|$, where S1 and S2 are the lengths of daughter sides (not the shared side) of two daughter cells. The value of the division section ranged from 0–0.2, meaning that the sides were divided through the middle. To quantify the relative position of the paired sides that were transected by cell divisions on the polygonal cells, two sets of interval sides between the paired sides were counted from the opposite directions at the point in time before division, and the smaller number was used in this study. More than 10 cells and divisions were examined for each thallus, and the results were expressed as mean $\pm$ SD.

# RESULTS

Conchospores are single cells that attach to the surfaces of glass culture bottles immediately after liberation. Conchospores first germinate into young thalli from meiosis after the formation of polarization (Fig. 1A). These young thalli contain only linearly ordered cells which were produced after sequential divisions in transverse orientation (Fig. 1B), and then expand on a two-dimensional plane to form a membranous sheet made of a single layer of cells. During early development, none of the cells was fully bordered by others (Figs. 1A–1D). As the thallus grows, more and more cells fully bordered by others (Fig. 1). Although thalli retain a rough lanceolate morphology, the cells at opposite positions on the same thallus can show different geometric patterns, different numbers of cells, or both (Figs. 1D and 1F). The entire developmental stages of *Pyropia haitanensis* thalli are shown in a diagram (Fig. 1G).

The number of sides of *P. haitanensis* thalli cells ranged from four to nine ($n = 6,434$ cells, Figs. 2A and 2B). Typically, cells with five sides were the most common during the initial stage of growth, but they become less as the thallus increases in length, supplanted by 6-sided cells (Fig. 2A). Regardless of the color of the thallus, 6-sided cells were the most prevalent in thalli longer than 0.08 cm, and 90% of the cells had 5–7 sides (Fig. 2B; Fig. S1). The distribution of angles was also closely related to thallus length. Cellular inner angles were in range of 45–176° ($n = 2,521$, Figs. 2C and 2D). The peaks of angle distributions shift from range of 90–110° to 110–130° along with the elongation of thallus, which was consistent with changes on average cell sides (Figs. 2C and 2D). The average numbers of cell sides in thalli shorter than 0.08 cm ranged in 5.0–5.69 and that of longer thalli were 5.72–5.97 (Fig. 2E). The average numbers of cell sides in thalli increased to almost 6 along with the growth of thallus. The average intracellular angles of thalli, which were longer than 0.08 cm, ranged in 112–119°; the average angles of shorter thalli were in range of 103–110° (Fig. 2F). Thus, when the average numbers of cell sides were increased to almost six, the average angles approached to approximately 120°, which is the average angle of hexagons. In addition, angles within the range of 100–140° accounted for >60% in thalli longer than 0.08 cm (Fig. 2D). The distributions and average values of these geometries suggested that cells favored regular polygons.

Cell divisions were studied through time-series observation. Most of the cells (93 ± 6%, a total of 187 cells in 8 thalli were examined) that before division were 5- to 7-sided cells (Fig. 3A). In the polygonal cells, cell divisions preferred to transect unconnected paired-sides (99%), and 97% of the paired-sides were intervened by 1 or 2 sides before division (Fig. 3B). In addition, most of the division sections (79%) were <0.2 and the average division section was 0.13 ± 0.03 (267 cells in 8 thalli were examined, Fig. 3C). These observations suggested that mitosis preferentially produced two daughter cells of the same size, this speculation was confirmed by the results of the current work that the size ratios of the smaller daughter cells to the larger ranged mainly (90%), between 0.7 and 1 and the average size ratio was 0.86 ± 0.03 (227 cells in nine thalli were examined, Fig. 3D).

Cell division always produced a new border shared by two daughter cells (Figs. 4A–4D), and they preferentially produced 3-fold junctions, which accounted for 99 ± 1%

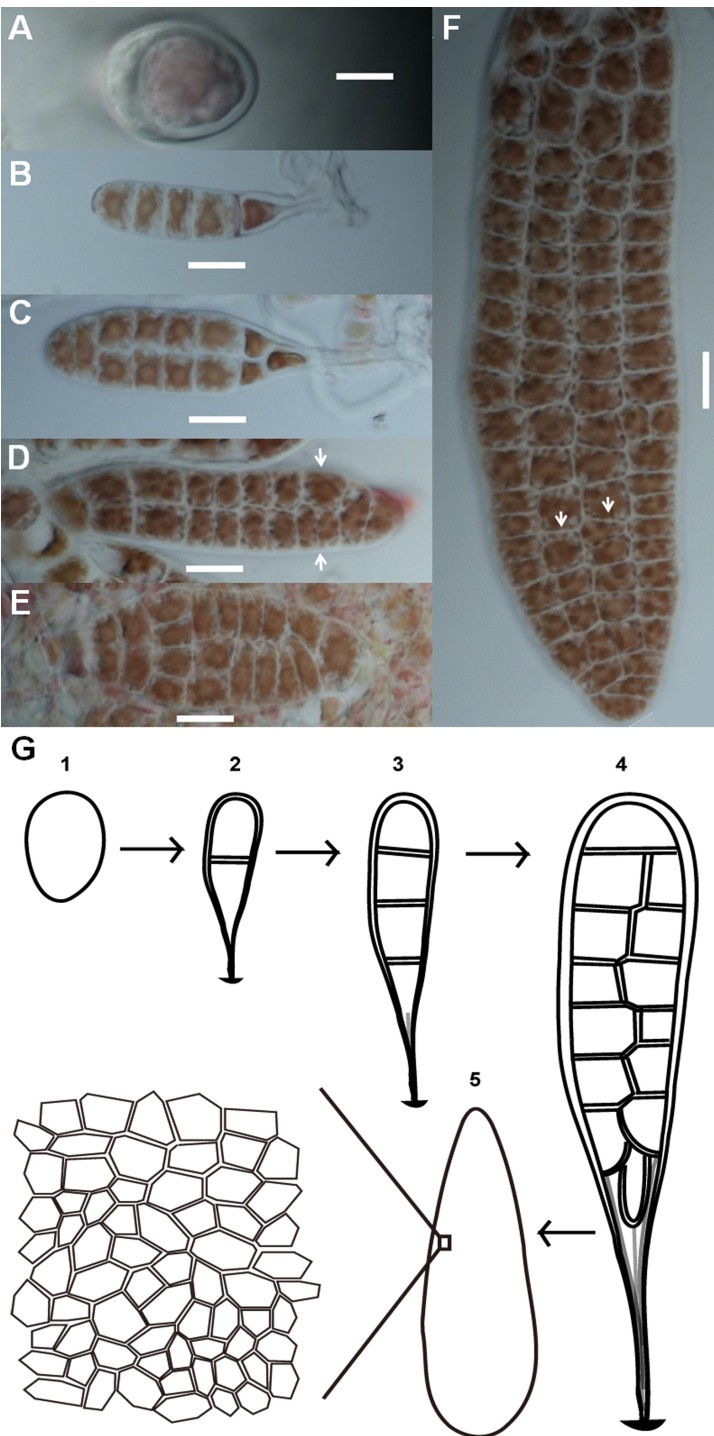

**Figure 1** **Morphogenesis of *P. haitanensis* during the thallus stage.** (A) Polarized conchospore. (B) Young thalli contained linearly ordered groups of five cells due to cell division alone along the apical-basal axis. The basal cells along the right side were elongated into rhizoids. (C–F) Cell proliferation of *P. haitanensis* thalli along a two-dimensional plane. White arrows showed different tessellation patterns at symmetrical positions on the same thallus. All scale bars are 20 μm. (G) All developmental stages.

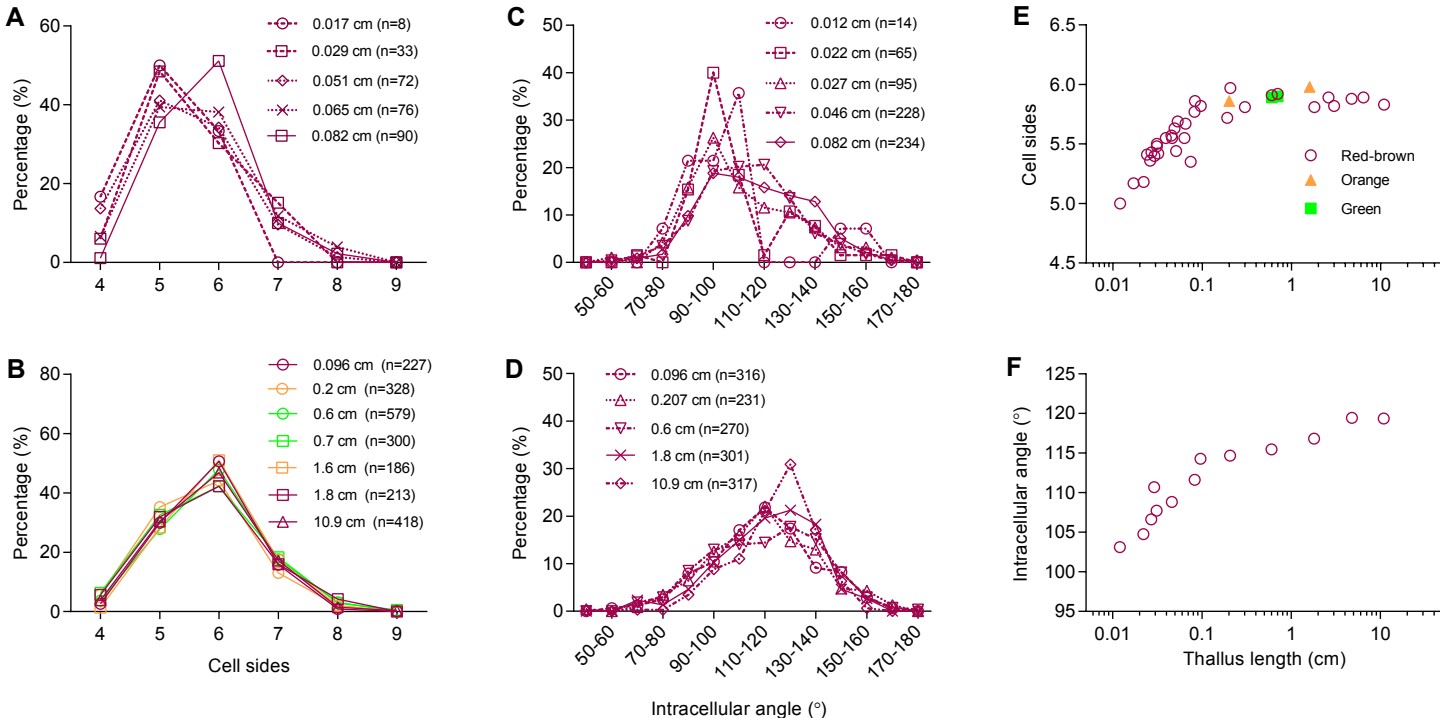

**Figure 2** **Relationship between thallus length and cell-packing geometries.** (A–D) Typical distributions of (A, B) number of cell sides and (C, D) intracellular angle of the thallus at different lengths; (E–F) average (E) number of cell sides and (F) intracellular angle of the thallus at different lengths. Symbols in red-brown, green, and orange represent the color of 3 strains of *P. haitanensis* Z-61, G-2, and O-9, respectively. Sample numbers are shown in parentheses.

of the total (782 junctions in five thalli were examined, Fig. 4E). Because the cells of *P. haitanensis* thalli are packed by rigid cell walls, the cell neighbor relationships remained stable until the topology was disturbed by the formation of two daughter cells by cell division (Figs. 4A–4D). Cell division was found to take place at night. Meanwhile, reorientation of new-junction-associated cell walls decreased the new angles formed in neighbor cells at a rate of 20 ± 3° per day over 2 days after the start of cell division (138 angles in 8 thalli were examined, Figs. 4A–4D and 4F).

## DISCUSSION

*Lewis (1926)* found that the average number of sides per polygonal cell to be close to six in cucumber skin. The core mechanism of this kind of phenomena was established by *Graustein (1931)*: the number of sides that meet at every vertex is always three. Then the tessellation with convex polygonal cells followed Euler's law to achieve an average of six sides. In this way, cell division must prefer to produce 3-fold junctions (vertexes) rather than junctions common to four or more polygonal cells. *Gibson et al. (2006)* also reported that the average number of sides of *Drosophila* wing cells remained close to 6, consistent with Euler's law, and they successfully used a Markov chain model validated with characteristics of cell division to predict the overall distribution of polygonal cell types.

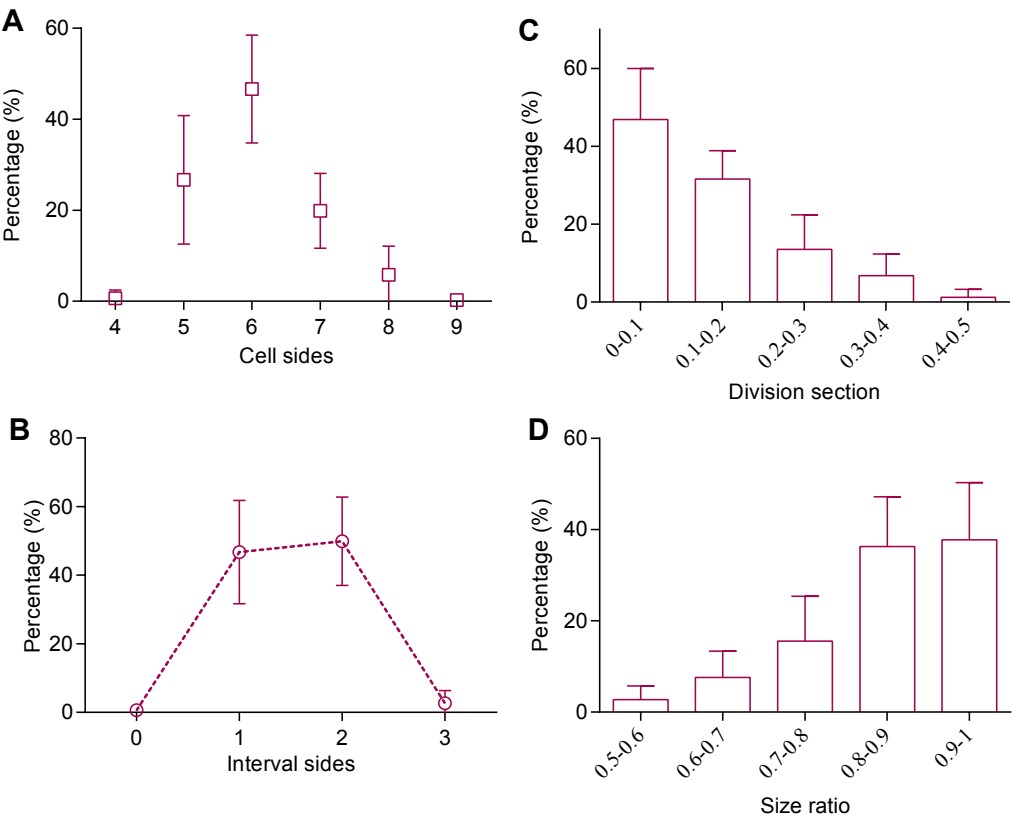

**Figure 3  Cell division cut at middle of selected paired-sides.** (A) Distribution of cell sides before cell division (187 cells in 8 thalli were examined). (B) Cell division favoring transection of unconnected paired sides (99%). The interval sides between paired sides indicates the relative positions on a polygonal cell before cell division, here the smaller of the two numbers enumerated from opposite directions. (C–D) Distributions of (C) division sections (267 cells in nine thalli were examined) and (D) size ratios of daughter cells (227 cells in nine thalli were examined). Division section indicates the relative distance between midpoint of a side and transection position on that side. The size ratios of daughter cells referred to the ratio of the small daughter cell to the big one. Sample numbers are shown in parentheses. The data are expressed as mean ± SD.

For this reason, they proposed that mechanisms of cell division could be mathematically sufficient to explain the predominance of 6-sided cells and the average six sides.

Results also showed the average number of sides of *P. haitanensis* cells increased to almost six as the thallus continued to grow, and the distribution of the number of sides in longer thalli remained very close to the measured and predicted values of *Drosophila* wings (Fig. 2B) (*Gibson et al., 2006*). Thus, the assumed conditions of Markov chain model are the major cause of mathematically achieved hexagonal cell patterns in multicellular organisms (*Gibson et al., 2006*). Based on previous studies and on the current results, the assumed conditions were reorganized as follows (*Gibson et al., 2006*; *Graustein, 1931*; *Lewis, 1926*; *Lewis, 1928*): 1. Cells bordered each other closely and grew uniformly as they proliferated on the flat plane; 2. Cell neighbor relationships were stable; 3. Cell division preferentially formed tricellular junctions; 4. Cell division favored equal division of cells by forming a new side shared by the two daughter cells.

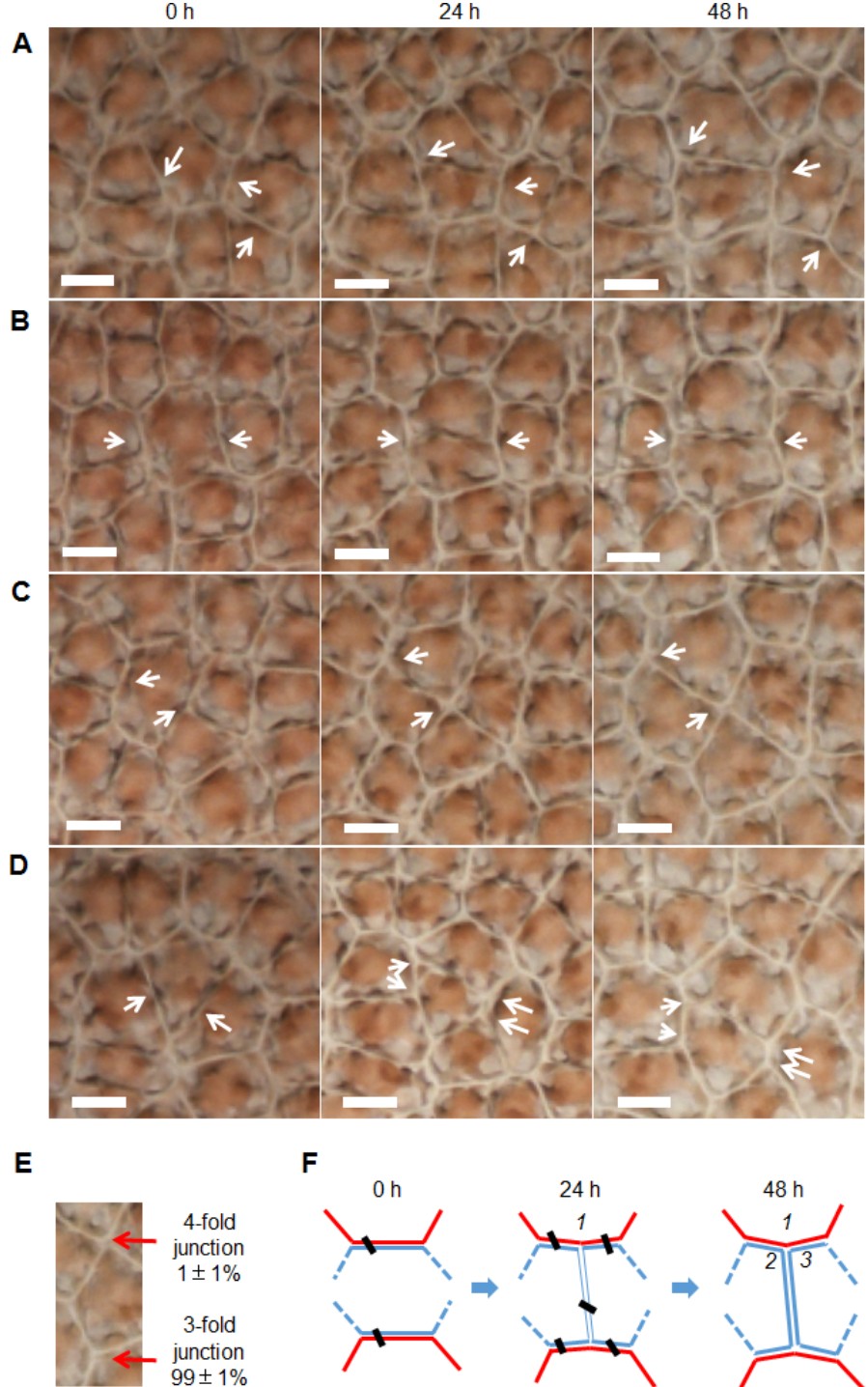

Figure 4 **Cell division associated reorientation of cell walls.** (A–D) White arrows point to the newly formed junctions and angles due to cell division. Due to reorientation, the new angles that formed in neighbor cells decreased during the first 48h after the beginning of cell division. In general, the reorientation remained for about two days until the boundary between two daughters became clear. All scale bars are 10 μm. (A–B) Cell division transecting unconnected two sides. 

**Figure 4 (…continued)**
(C) Cell division transecting a side and a vertex of an 8-sided cell, which produced a 4-fold junction. (D) Cell division transecting connected two sides. Arrows indicate where the daughter cell gained two sides from dividing neighbor cells, which prevented the creation of triangular cell. (E) About $99 \pm 1\%$ of junctions were 3-fold and the rest were 4-fold. (F) Diagram of cell division of *P. haitanensis* thallus. A cell in blue color with $\geq 4$ sides, its paired-sides which shared with two red cells would be transected by cell division. Cell division increased the number of cells by one and the number of counted cell sides by six. The reorientation of cell walls around a newly formed junction led to adjustments of one angle (1) in neighboring cells and two angles (2 and 3) in the daughter cells. During the first two days, the rate of decrease of the new angles in neighbor cells was $20 \pm 3°$ per day (angle 1, 138 angles in 8 thalli were examined). The data are expressed as mean $\pm$ SD.

Conditions 1–2 were supported by previous studies and the current observations (*Gibson et al., 2006*; *Graustein, 1931*; *Lewis, 1926*; *Lewis, 1928*). Cell rearrangement has been proposed as a primary mechanism to control the formation of cell hexagonal packing in *Drosophila* wings, which resulted in exchanges of neighbor cells (*Classen et al., 2005*; *Farhadifar et al., 2007*; *Heller et al., 2016*). *Gibson et al. (2006)* did not detect cell rearrangement in proliferating *Drosophila* wing cells. However, a recent study including automated, systematic high-throughput analysis methods showed 13 times of cell rearrangement per 1,000 cells per hour (*Heller et al., 2016*). The cells of *P. haitanensis* thallus are packed by rigid cell walls the same way as the plant cells, which indicates that each walled cell and its neighbor cells adhere to each other to prevent cell migration (*Taiz & Zeiger, 2010*). The assumed condition 2 is suitable to the current study on the thalli of the macroalgae *P. haitanensis* and previous studies on the higher plant cucumber (*Graustein, 1931*; *Lewis, 1926*; *Lewis, 1928*).

Because of the stable cell neighbor relationships (e.g., Fig. 4), the formation of tricellular junctions was found to depend on the cut position of the divide during cell division: 1. Cell division always cut through two sides of a cell rather than two vertexes or one vertex and one side. In this study, only <1% of observed cells were divided through both a vertex and a side (Fig. 4C), and none were cut through two vertexes of a polygonal cell. 2. Cell division favored division at different positions to produce tricellular junctions. That is supported by our observations that only 3- and 4-fold junctions were identified and tricellular junctions accounting for 99% (Fig. 4E).

Under conditions 1–3, the paired sides of mother cells were found to split into four sides which it shared with neighboring cells, while a new side would form, shared by two daughter cells (Figs. 4A–4D). This increased the number of cells by one due to the division, but the net increase of shared sides was three. In this way, every division increased the number of sides by six. The average number of cell sides would then approached six because of the large number of cell divisions taking place during the growth of the thallus (Fig. 2E). For the same reason, the average number of sides per cell would be closes to three if cell division cut through vertexes or closes to four if cell divisions favored producing 4-fold junctions. As for the non-walled cells of epithelia, the assumptions used in Markov chain model (*Gibson et al., 2006*) are challenged by the observations of cell rearrangements (*Classen et al., 2005*; *Heller et al., 2016*). However, whatever cell rearrangements may be ignored or emphasized, the overall distribution of polygonal cell in epithelial tissues (*Classen et al., 2005*;

*Gibson et al., 2006*; *Heller et al., 2016*) is very close to walled cells in the present study and previous studies (*Graustein, 1931*; *Lewis, 1926*; *Lewis, 1928*). Because the cell rearrangements resulted in cells with a larger number of sides that lose an edge to cells with a smaller number of sides (*Heller et al., 2016*), it is possible that cell rearrangements would not change the distribution of polygonal cells of the epithelial tissue on a global scale, but further tests are needed to confirm this hypothesis.

However, conditions 1–3 were not found to necessarily determine the distributions of polygonal cells. The Markov chain model was found to predict the distributions of 5- to 7-sided cells very well by leaving out 3- and 4-sided cells due to their low prevalence ($\approx$0 and $\approx$4%, respectively) upon real observation (*Gibson et al., 2006*). If the orientation of cleavage was random, then the rate at which two connected sides (0 interval side) would transect by cell division would be roughly 60%, 43%, 35%, and 30% for 4-, 5-, 6-, and 7-sided cells, respectively. However, in both *Drosophila* wing cells (*Classen et al., 2005*; *Farhadifar et al., 2007*; *Gibson et al., 2006*), cucumber skin cells (*Lewis, 1926*; *Lewis, 1928*), and longer *P. haitanensis* thalli, the prevalence of 4-sided polygonal cells was always <5%. Even though cells would gain an average of one side each time from its neighbors divided (*Gibson et al., 2006*), a larger proportion of 4-sided cells would still be expected to be found under real conditions. In this way, the measured values contradict the assumption that cell division would be randomly orientated (*Gibson et al., 2006*).

Biological processes and mechanisms are also more complex than had once been thought. It seems to make sense that cells would divide equally during mitosis. Theoretically, if cell division prefers to divide 5–7-sided regular polygons in half, then the cleavage plane should always cut unconnected paired sides. The present study confirmed this speculation (Fig. 3B). The current study is consistent with a previous model study by *Patel et al. (2009)*. They found different cleavage patterns generated different distributions of polygonal cells, and their results suggested that strong divisional symmetry is required to reproduce the predominance of hexagonal cells. In addition, cell division was found to favor cutting in the middle of the sides (Fig. 3C). In this way, to achieve the equal division, both the relative position of paired sides that were cut by cell division and the points at which they were cut were selected (Fig. 3D). This shows that the reason for cell sides $\geq$4 was that mitosis favored equal division of cells, which could help improve the assumptions of the Markov chain model (*Gibson et al., 2006*). Moreover, 0.6% of paired sides (187 cells in eight thalli were examined) were connected and transected by cell division, which was expected to take place in 0.6% of triangular cells. However, all of the examined cells ($n = 6{,}432$) had 4 sides or more. The creation of triangular cells was prevented by gain of sides from divisions of neighbor cells (Fig. 4D).

It must be emphasized the number 6 is just a statistical average. The number of sides of an individual cell is not a fixed number, but rather fell mainly within the range of 4–9 (Fig. 2). The increase in the average number of cell sides (Fig. 2E) reflected the kinetic geometry of all the polygonal cells on a global scale during the development of *P. haitanensis* thalli. Each cell in the tissue needed to cooperate or deal with other cells to help maintain appropriate geometry. For example, in the 2D tessellation, the sum of angles around a multi-cellular junction must equal 360°. Furthermore, the mathematical laws of 2D tessellation suggested

that the geometry of individual cells is not random, especially when the tessellation was characterized by specific distributions of polygonal cells and the average number of sides approached 6 as the number of cells increased (*Aboav, 1980*; *Grünbaum & Shephard, 1987*; *Lewis, 1928*; *Lord, 2016*; *Weaire & Rivier, 1984*). Therefore, if the geometry of individual cells were random or controlled only by each cell itself, then it would be quite possible that a group of cells would not tile a flat plane. However, during this study, healthy thalli of *P. haitanensis* did not contain holes or gaps between cells. In addition, if intracellular angles remain stable, then newly formed angles would be determined solely through cell division, which means that the new angles of the two daughter cells should have equal chances to be either ≤90° or ≥90°, and angles of 180° would be formed inside neighbor cells. However, the majority of inner angles were within the range of 100–140° when cell sides ranged in 5.72–5.97, and only minor amount of angles were observed <60° or >170° (Figs. 2B and 2D). These findings could not be explained by cell division alone.

Newly formed tricellular junctions were located exactly at the cut points on paired sides because of cell division (Fig. 4). Time series observation indicated that sides were reoriented around each newly formed junction, resulting in adjustments of angles which was associated with these sides, and the newly formed angles within neighbor cells showed a decrease rate of 20° per day during cell division. In this way, cell division and reorientation of sides were the primary reasons for the distribution of intracellular angles (Figs. 2 and 3). The adjustments of intracellular angles indicated that cells of *P. haitanensis* thalli cooperate with each other by favored the formation of regular polygonal patterns. The cell walls in regular polygonal patterns could render the shape of protoplasm more spherical, which could decrease the surface energy (*Chen, 2008*; *Ingber, Wang & Stamenovic, 2014*). Thus, the reorientation may have been mediated or controlled by biophysical properties.

Recently, by use of Voronoi diagrams, a surprising study by *Sanchez-Gutierrez et al. (2016)* proposed a new mechanism that, independent of cell proliferation mechanisms, is through physical cellular constraints drives the distribution of polygonal cells. Each convex polygonal cell is a Voronoi cell, generated from a specified point (called a seed). Each such cell is a region consisting of all points closer to its seed than to any other seed. The current study showed that cell division cut through the middle of the selected paired sides to produce equal-sized daughter cells, which means our findings could offer constraints for using Voronoi diagrams to mimic the cellular proliferation processes by locating the positions of seeds in daughter polygonal cells. Voronoi tessellation has an obvious advantage over cell proliferation mechanisms in that the former could explain the conserved polygon distribution in non-proliferative tissue (*Gibson et al., 2006*; *Graustein, 1931*; *Patel et al., 2009*; *Sanchez-Gutierrez et al., 2016*; and the current study). The differences and connections between the mechanisms dependent and independent of cell proliferation require further study.

## CONCLUSION

The evolution of multi-celled organisms was a tremendous step in the early history of life, but the mechanisms by which the cells are arranged to form organisms with specific

morphology are still unclear. *P. haitanensis* inhabits the intertidal zone which is subject to fast and periodic changes in temperature, seawater, light, and nutrient availability. Thus, this macroalgae species needs to tightly, precisely, and effectively control their growth and cell division to maintain their typical morphological features. The current study indicated that cell division mainly involved cutting through middle of selected paired sides to mathematically bring the average number of sides per cell increased to almost 6 as the thallus elongated; cellular angles changed through reorientation of cell sides, which, with cell division, determined the distribution of angles. These results offer some help in understanding that the physiological and geometric performances of a single cell exhibit a global pattern in a multicellular organism.

### Funding

This work was supported by the National Natural Science Foundation of China (Grant No: 41276177), and the Natural Science Foundation of Fujian, China (Grant Nos: 2014J07006 and 2014J05041). The funders had no role in study design, data collection and analysis, decision to publish, or preparation of the manuscript.

### Grant Disclosures

The following grant information was disclosed by the authors:
National Natural Science Foundation of China: 41276177.
Natural Science Foundation of Fujian: 2014J07006, 2014J05041.

### Competing Interests

The authors declare there are no competing interests.

### Author Contributions

- Kai Xu and Chaotian Xie conceived and designed the experiments, performed the experiments, analyzed the data, contributed reagents/materials/analysis tools, wrote the paper, prepared figures and/or tables, reviewed drafts of the paper.
- Yan Xu, Dehua Ji, Ting Chen and Changsheng Chen performed the experiments, contributed reagents/materials/analysis tools, reviewed drafts of the paper.

### Data Availability

The raw data has been supplied as a Supplementary File.

### Supplemental Information

Supplemental information for this article can be found online at http://dx.doi.org/10.7717/peerj.3314#supplemental-information.

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
