# Peer review of "Cells tile a flat plane by controlling geometries during morphogenesis of Pyropia thalli"

_PeerJ, doi:10.7717/peerj.3314_

## Round 0.1 · original submission · Major Revisions

· Academic Editor

Major Revisions

The paper is interesting although needs to be improved.. In particular, materials and methods with statistics and the English have to be revised.

·

Basic reporting

This work brings a very interesting model to the analysis of planar geometry in packed tissues. The reviewer agrees with the author that the characteristics of Pyropia haitanensis thalli are excellent to the analysis of this primary problem. The data is clear and well presented although it is possible to improve some aspects (see below). The work provides interesting data about the organization of this red algae and how it compares with classical analyses on epithelial organization.

However there are some important points also to be corrected before the paper can be considered for publication. In particular I consider the point number 6 very relevant.

Experimental design

No comments

Validity of the findings

I find some problems with the data since I miss the presence of error bars.

I also find a problem with the main conclusion.

Please see below for more details.

Additional comments

1- I consider the introduction incomplete.

- There are some essential works that should be introduced when talking about polygonal geometry and planes. Starting by the works of Lewis and Aboav-Weaire on some general laws that affect these types of tissues.

- The polygon distribution concept should be also introduced.

- Other recent work has shed light on the mechanism that drive packed tissue organization. Sanchez-Gutierrez et al. (EMBO J. 2016) described that there is a physical constraint that makes all packed tissues to pack similarly to Voronoi diagrams (Voronoi, 1908). This is extremely relevant for this work, since this physical constraint is affecting how cells pack in proliferating but also non-proliferating tissues. This indicates that cell division and cell rearrangements are subject to these restrictions, so they should be considered.

- Related to the importance of cell division angles is also very surprising that the paper from Patel et al. (Plos Comp. Biol 2009) is not mentioned. This should be included in the introduction and discussed in the Discussion section.


2- Regarding the data presented in the figures:

- Figure 2 A and C should include all the X-axis labels. Legend in 2B must be ordered (similarly to A, C or D).

- Figure 3 and Figure 4 show data from different groups of cells. How many different thalli where used to obtain the data from these cells? The figures would be greatly improved if it is shown the average data for each thalli analyzed. This will enable the usage of error bars.


3 - Line 121. Sentence looks incomplete. What is the meaning? Please clarify.


4 - In the discussion section the first sentence is not correct. The average number of sides of the cells is close to six since they are convex polygons in a tessellation. It is Euler´s law what dictates that a tessellation with convex polygon leaving no empty space should have an average side number of six. This is independent of proliferation (this is supported by the data of Sanchez-Gutierrez et al. , EMBO J. 2016).

5- In line 152 authors declare that cell rearrangements were not observed in Drosophila wing cells. However, a recent paper performing live imaging and tracking of individual cells showed that assumption is wrong. This sentence should be corrected and the paper cited: Heller et al. Dev Cell 2016.

6- I found the paragraph between lines 196-202 is speculative and unclear. This is an important part of the manuscript since supports the main conclusion of the abstract. It should be greatly improved and changed to try to explain how cells “cooperate” to do not leave gaps between them

This is an essential point since it is related to the main conclusion in the abstract. Therefore it must be very clear and convincing.

In addition, how it is related to, for example the case of epithelia, where there is not cell walls, but strong adhesion between cells? This should be discussed in this section, since the manuscript is full of references to the epithelial geometry.

7- In line 217 biophysical properties are mentioned as responsible for reorientation of cells. What are they coming from? This also should be discussed further since it is a relevant possibility that explain tissue organization. Again this can be related to Voronoi diagrams as described Sanchez-Gutierrez et al. (EMBO J. 2016).

8- In general the authors should greatly improve the introduction and the state of the art of the field that they want to study.

Reviewer 2 ·

Basic reporting

An editorial revision by native English speaking colleagues or editing services is very necessary to ensure the quality of the paper.
Some definitions and statements are not clearly presented. For example, in Line 51-54: The authors should define what is the ‘intracellular angle’ ? No definitions and related references are found either in the manuscript or in the two cited references.
In addition, authors should provide concise but sufficient figure legends for each figures. For example, it is difficult to understand the charts without labels for X-axis in Fig.2A and Fig. 2C. Authors should clarify the relations between Fig.2A and 2B, while Fig. 2C and 2C.
To make the major conclusion that appropriate cell-packing geometries maintained by cell division and reorientation of cell walls, the analysis processes presented in Fig.4 should be moved to the Results instead in the Discussion section. Supplemental Fig. 2 is a key result to support above statement and should be presented as a major figure.

Experimental design

Part of Materials and Methods is needed to be carefully rewritten. For example, in Line 78, whether the side lengths that were larger than the thickness of the jointed cell wall between two adjacent cells were considered?
Additional analysis is needed to fully understand the morphogenesis of Pyropia haitanensis thalli. Authors nicely demonstrated the different developmental stages of young thalli in Fig.1. The analysis of cell division and cell shape are important data to understand the cell proliferation since these initial stages, such as, whether the linearly ordered cells were produced after sequential divisions in transverse orientation (Fig. 1B).
A diagram of the entire developmental stages of Pyropia haitanensis thalli in Fig.1 will be appreciated by the readers.

Validity of the findings

In this study, Xu and coauthors showed analysis data about the cell-packing geometries of Pyropia haitanensis thalli, which are multi-celled organisms consisting of a single layer of polygonal cells. Authors followed the cell proliferation process of thalli at different growth stages and found that the cell-packing geometries in Pyropia haitanensis thalli are maintained by cell division and reorientation of cell walls. The manuscript could be of interest to the readers of PeerJ if the paper quality were improved after a careful revision.

Authors need to be careful making statements that are not supported by the represented data. Such as, in Line 99-101, ‘For example, the two cells indicated by the lower arrow in Fig.1D…which was because of different division times’. Authors should give the evidences to prove whether these two cells were generated by an excessive division in the lower row comparing the cell sitting in the upper row.
In Line101-103, ‘which indicates cell divisions transected the cell at irregular directions’. But, the cells are well-organized shown in Fig.1F, a pattern that is produced by combination of transverse and longitudinal divisions referring to the growth axis instead of division at ‘irregular directions’.

---

## Round 0.2 · Minor Revisions

· Academic Editor

Minor Revisions

The manuscript should be modified according to the final comments provided by reviewers.

·

Basic reporting

Good.

Experimental design

Good.

Validity of the findings

Good.

Additional comments

I consider that the authors have improved the manuscript according to the suggestions of the reviewer.
I find it suitable for publication now.

My only last suggestion is include a comment in the introduction when they comment about the paper from Sánchez-Gutiérrez et al. 2016.

They say:
"A recent work by Sánchez-Gutiérrez et al. (2016) has shed light on the mechanism that drives packed tissue organization. They found that a physical constraint that induced by the balance of forces between cells in tissues can cause all packed tissues to be arranged in a manner similar to Voronoi diagrams. (Page 4, Lines 67–70)."

I think that it should be clarify that this paper also demonstrated that a packed tissue can be enriched in hexagons without the need of proliferation mechanisms (this is the case for muscles). This is closely related to the comment:

"The current study is consistent with a previous model study by Patel et al. (2009). They found different cleavage patterns generated different distributions of polygonal cells, and their results suggested that strong divisional symmetry is required to reproduce the predominance of hexagonal cells. (Page 9, Lines 235–238)."

Again, Sánchez-Gutiérrez et al. showed that it is possible the predominance of hexagons without cell proliferation.

I also want to thank the author for the answer in the point 4, and the reference to the paper by Graustein (1931).

Reviewer 2 ·

Basic reporting

no comment

Experimental design

no comment

Validity of the findings

no comment

Additional comments

The authors addressed my major concerns and I am satisfied with the new version.
One minor point again about the labeling and description for Figures, such as, the Red-brown, Orange, Green in Fig. 2E. According to the Materials and Methods, are those come from different strains? Authors need to make this clear.

---

## Round 0.3 · accepted · Accept

· Academic Editor

Accept

Thank you for improving your work and congratulations again.